# Polyglycerol Sebacate Elastomer: A Critical Overview of Synthetic Methods and Characterisation Techniques

**DOI:** 10.3390/polym16101405

**Published:** 2024-05-15

**Authors:** Mariella Rosalia, Davide Rubes, Massimo Serra, Ida Genta, Rossella Dorati, Bice Conti

**Affiliations:** Department of Drug Science, University of Pavia, Via Taramelli 12, 27100 Pavia, Italy; davide.rubes01@universitadipavia.it (D.R.); serra@unipv.it (M.S.); ida.genta@unipv.it (I.G.); rossella.dorati@unipv.it (R.D.); bice.conti@unipv.it (B.C.)

**Keywords:** polyglycerol sebacate, synthesis, characterisation, polymer tailoring

## Abstract

Poly (glycerol sebacate) is a widely studied elastomeric copolymer obtained from the polycondensation of two bioresorbable monomers, glycerol and sebacic acid. Due to its biocompatibility and the possibility to tailor its biodegradability rate and mechanical properties, PGS has gained lots of interest in the last two decades, especially in the soft tissue engineering field. Different synthetic approaches have been proposed, ranging from classic thermal polyesterification and curing to microwave-assisted organic synthesis, UV crosslinking and enzymatic catalysis. Each technique, characterized by its advantages and disadvantages, can be tailored by controlling the crosslinking density, which depends on specific synthetic parameters. In this work, classic and alternative synthetic methods, as well as characterisation and tailoring techniques, are critically reviewed with the aim to provide a valuable tool for the reproducible and customized production of PGS for tissue engineering applications.

## 1. Introduction

Soft tissue engineering consists of the use of 3D scaffolds to support cell attachment, proliferation and differentiation, providing cues to promote tissue repair and regeneration [1,2]. “Soft” is a very generic term and refers to almost all types of tissue, including skin, muscular, cardiovascular, gastrointestinal and nervous tissue, but excluding, for example, bone and blood [3]. Besides the specific characteristics of each soft tissue type, due to their peculiar functions [4,5], several common requirements need to be addressed in the design of tissue engineering scaffolds, including chemical and biochemical properties, geometry and architecture, mechanical characteristics and porosity [6,7]. Taken together, all these aspects contribute to define the biocompatibility and success of the implanted tissue engineering scaffold. Biomaterials play a key role because they are the main components of tissue engineering constructs and are generally designed to mimic extracellular matrix (ECM) building blocks. Soft tissue extracellular matrix is composed of a high amount of fibrillar proteins with a structural function, namely collagen and elastin [8,9]. Collagen is, by definition, the scaffolding protein of the human body, organized in a rope-like structure composed of several hierarchically organized fibres [10]. Elastin is the major protein in extensible tissues and is composed of covalently linked soluble precursors that form fibril-like aggregates [11]. Both macromolecules confer elasticity to soft tissue thanks to the presence of crosslinks between their structural units; the enzyme lysil oxidase catalyses the formation of aldol bonds between lysine (or hydroxylysine) residues on adjacent triple-stranded procollagen helixes or tropoelastin chains. The number and density of crosslinks determine the degree of elasticity and the resilience of each protein subtype [10,11]. Depending on the tissue type, the composition and organisation of ECM varies, and the amount of collagen and elastin is related to tissue function. According to the need for matching the mechanical properties of native tissue, the development and use of elastomeric biomaterials has gained lots of interest in tissue engineering [3]. Elastomers are polymeric materials characterized by a high resilience, i.e., the ability to recover their original shape after big deformations, without any rupture. This feature is given by the presence of crosslinks between adjacent polymeric chains, as in collagen and elastin. According to the type of intermolecular bonds, elastomers can be classified as thermoplastic elastomers, when physical, weak forces (hydrogen bonds, dipolar forces, hydrophobic forces) form the crosslinks, or thermoset elastomers (or simply elastomers), when covalent intermolecular bonds are present [8]. Several synthetic, biodegradable elastomers have been reported in the literature [12], and among them poly (glycerol sebacate) (PGS) have gained lots of interest in the last 20+ years, as shown by the increasing number of publications reported in the Web of Sciences (Figure 1).

PGS is a thermoset elastomeric biopolymer, first synthetized by Robert Langer and co-workers at MIT in 2002 [13], and obtained through the polyesterification of glycerol and sebacic acid. Both monomers are endogenous molecules found in the human body. Glycerol is a triol and a fundamental component of fats, such as triglycerides and phospholipids, and can also take part in glycolysis and gluconeogenesis [14]; sebacic acid is a dicarboxylic long chain acid (C10) that can be transformed into acetyl-CoA and takes part in the Krebs cycle [15]. Hence, after the hydrolysis of the ester bond, the two non-toxic monomers can be easily metabolized by the organism. Moreover, FDA already approved glycerol-containing and certain sebacic acid-containing polymers for medical use [13]. PGS was widely studied and tailored for applications in soft tissue engineering, including cardiac tissue [16,17,18,19,20], vascular tissue [21,22,23,24], nervous tissue [25,26,27,28], skin [29,30,31,32] and cartilage [33,34,35]. Biodegradation studies performed in vitro and in vivo demonstrated surface erosion and minimal formation of fibrotic capsule, with complete absorption of the graft within 60 days [13,27,36]. Scaffolds with tailored mechanical properties were obtained, with Young moduli ranging from 0.03 to 1.4 MPa, elongation at break from 125% to 265% and ultimate tensile strength higher than 0.2 MPa [13,37,38]. Cytocompatibility was proven on different cell lines, including fibroblasts, endothelial cells, vascular smooth muscle myocytes, cardiomyocytes, and Schwan cells [27,31,39]. Starting from this background, PGS can be considered an attractive polymer for tissue engineering approaches. While PGS’ biocompatibility, biodegradability, and tuneable mechanical properties make it an interesting biopolymer, its clinical application still represents a challenge, mainly due to its limited processing and sterilization options, and unknown long-term efficacy and safety, which still need in-depth evaluation [8]. On the other hand, these challenges also represent a starting point for carrying out further research on PGS-based biopolymers, as witnessed by the constant research efforts on this material. PGS and its derivatives and composites were already recently reviewed by other authors, focusing on their processing techniques and application for tissue engineering purposes [40,41,42]. This review intends to critically overview established and newer synthetic methods, discuss their possible optimization and report the most common polymer characterisation techniques. The aim is to provide a support for the standardised and sustainable production of PGS for tissue engineering purposes.

## 2. Classic Synthesis 

### Thermal Polycondensation

Poly(glycerol sebacate) is a crosslinked polyester, synthetised through polycondensation between glycerol hydroxyl groups and sebacic acid carboxylic functions [13]. The reaction can be divided in two steps; first, there is the pre-polymerisation phase with the formation of single polymer chains, and subsequently, the curing phase, in which crosslinks are formed.

The pre-polymerisation of sebacic acid and glycerol follows a step-growth process [43]; the reaction initially takes place between monomers, but rapidly involves just synthesised dimers, trimers, and other oligomers because the reaction can take place at multiple sites. The monomers are consumed very fast so that, at an average polymerisation degree of 10, less than 1% by weight of monomers is left, whereas the overall elongation of the polymeric chain occurs slowly [9]. The pre-polymerisation reaction between glycerol and sebacic acid and the chemical structures of the obtained pre-PGS species are shown in Figure 1 and since it is a condensation reaction, water is produced as a by-product. No secondary reactions were observed.

The most reactive glycerol hydroxyl groups involved in the pre-polymerisation step are the primary ones in position 1 and 3; therefore, 1-monoacylglycerides and diesters 1,3-diacylglycerides are formed. The formation of 2-monoacylglycerides and 1,2-diacylglyceride is less favoured because of the lower reactivity of secondary hydroxyl groups. When glycerol availability decreases due to reactant consumption or evaporation, the formation of branched molecular species, among which 1,2,3-triacylglycerides have been reported, occurs [37]. Some authors have also reported the formation of cyclic structures due to intra-molecular condensation reactions that occur when low reactant concentration is used [44]. During the polymerisation step, the growth of branching chains is responsible for an increase in the system’s viscosity. Before the system turns viscous, the reaction is said to be in the kinetic regime, in which the conversion rate is fast thanks to the high mobility of the involved species. In this first part of the process, the reaction is controlled by a first-order kinetic, in which the reaction rate depends only on the concentration of the unreacted monomers. Afterwards, the collision frequency between reactive moieties decreases and the reaction becomes diffusion driven; in this second part of the process, a sharp increase in molecular weight can be observed [45]. The main drawbacks of this type of polymerisation method are the slow rate and random formation of chains, a high polydispersity index and the formation of moderately long polymeric chains, generally under 100 kDa [9].

In the curing step, crosslinking between PGS pre-polymer (pre-PGS) chains is performed thanks to the selection of a trifunctional monomer, glycerol [13]. While in the first pre-polymerisation step, the two primary alcoholic groups in glycerol are involved, only the secondary alcoholic groups are free to further react during the curing procedure [8]; the transesterification of random coiled pre-PGS molecules occurs between branches, locking the polymer chains into a 3D network (Figure 2) [46]. The advantage of using the same covalent bonding in crosslinks as in the main polymeric chain makes it possible to avoid heterogeneous polymer degradation and reduce excessive crosslinking, which would lead to less resilient, more brittle material [13].

Several reaction conditions have been applied in both the pre-polymerisation and curing steps. In the first pre-polymerisation reaction, glycerol and sebacic acid are usually mixed at a 1:1 molar ratio [13,16,36,37,39,47,48,49], even if also different ratios of monomers are reported in the literature, including 1:2, 2:1, 2:3 and 2:5, 3:2 and 3:4, 4:3, 5:4 and 5:6 of glycerol and sebacic acid [33,43,44,50,51,52]. The solvent-free monomer mixture is then heated between 110 and 130 °C for 24 to 120 h [36,37,39,47,48] at atmospheric pressure and under inert atmosphere (argon or nitrogen) in order to avoid the excessive loss of volatile glycerol that occurs by applying a vacuum [37] and the oxidation of reactants [8], respectively. When shorter reaction time is applied, additional heating can be performed under vacuum conditions, at 10 to 50 mTorr and for a period of 5 to 50 h. During pre-polymerisation, the water-side product can be removed using a cooled condenser or a Dean–Stark trap [46]. This first synthetic step is usually performed in neat condition, without any solvent; the use of 40% toluene to avoid reactant overheating is also reported in the literature [43]. For the second step, the curing, pre-PGS is heated between 120 °C and 150 °C for 24 to 96 h under vacuum (10–100 mTorr) conditions. High temperatures, long reaction time and vacuum are needed to enhance secondary hydroxyl reactivity and promote a crosslinking reaction [13,39,46,48,49]. After curing, the polymer is no longer soluble in organic solvents, including ethanol, isopropanol, N,N-dimethylformamide, dioxane, and tetrahydrofuran (THF) [13].

## 3. Alternative Synthesis

The classic synthetic route for PGS is particularly energy consuming due to the high temperatures and the prolonged reaction time for both polycondensation and curing steps [53]. Moreover, a high amount of purge gas and vacuum conditions are also required to accelerate the reaction, leading to the loss of glycerol during water by-product evaporation, modifying the reactant ratio and chemical composition of the final product [54]. Hence, alternative synthetic strategies have been proposed (Figure 2), including microwave-assisted polycondensation [53,54,55,56,57], photocuring [38,58,59,60] and enzymatic synthesis [61,62,63].

### 3.1. Microwave-Assisted Polycondensation

Nowadays, microwave-assisted organic synthesis (MAOS) is routinely used due to its ability to speed up and enable reactions that would otherwise be time- and resource-consuming, including isomerization, condensation, cyclization, reduction, and coupling reactions. Microwaves (Mws) are non-ionizing radiations, with a wavelength between 1 mm and 30,000 mm and a frequency from 300 GHz to 1 GHz, that do not affect the molecular structure of compounds. The electromagnetic field generated by Mws induces dipolar oscillation and ionic conduction, which aligns dipolar species and ionic particles to perpetual reorientation cycles, leading to almost instantaneous, non-contact heating with minimal energy dispersion [64,65]. The solvent overheating and the related high pressures that can be obtained by working in closed-reaction vessels that allow, in most cases, to accelerate organic reactions and reduce operation time, solvent loss, and the formation of unwanted side-products. Moreover, selective heating can be achieved using chemical species with an appropriate loss tangent, which results in a more effective and site-specific coupling to Mws and in the possibility to perform solvent-free reactions. The possibility to obtain purer products and high yields of production in a fast and environmentally friendly manner made MAOS particularly attractive in polymer synthesis [64,66].

Aydin and co-workers firstly worked on the Mws synthesis of pre-PGS [53]; glycerol and sebacic acid were heated intermittently for 3 min in a domestic microwave oven set at 650 W, with 10 seconds venting pauses to avoid overheating, the build-up of gas, and to remove water vapor. Further curing was performed conventionally in a vacuum oven. The FTIR spectra of Mws pre-PGS showed differences compared with classic synthetised pre-PGS; the peaks at 1159 and 1735 cm^−1^, corresponding to C-O and C=O bonds, were not similar, whereas crosslinked PGS polymers FTIR spectra did not show differences. Indeed, 1H-NMR spectra confirmed the synthesis of pre-PGS using the Mws method, and the differences reported by FTIR and DSC spectra were related to an important glycerol loss during microwaving, which occurred due to the use of an open reaction vessel. In fact, the final monomer ratio in Mws PGS was 78:22 for sebacic acid and glycerol, respectively, instead of the pristine 1:1 reactant mixture. An alteration of the monomer ratio can lead to unexpected mechanical properties and should, therefore, always be evaluated.

The effect of glycerol loss during microwaving on the final product characteristics was further investigate by Li and co-workers, who concluded that the variation of glycerol/sebacic acid ratio is caused by a high amount of unreacted glycerol after MW pre-polymerisation, which evaporates quickly during curing in a vacuum oven. The glycerol loss influences the degree of esterification (DE) and the boundaries between the different physical states of the polymer (e.g., liquid to soft elastomer); therefore, an accurate determination of DE should be used to compare classic and MW-synthetised PGS, rather than the reaction time or temperature. In fact, if FTIR spectra and the mechanical analysis of polymer samples obtained with different synthetic methods, but with the same DE, were compared, similar results would be obtained [54].

In agreement with the importance of controlling and tuning the DE of PGS, Lau et al. proposed a new approach to efficiently use Mws irradiation for the consistent and controlled synthesis of pre-PGS; glycerol and sebacic acid were mixed in a 1:1 ratio in toluene, and concentrated sulfuric acid was added as the catalyst. Toluene (boiling point of 110 °C) was chosen as a solvent to avoid excessive heating and protect glycerol (boiling point of 290 °C) from evaporation. Indeed, in the adopted conditions, the maximum temperature reachable by the reaction was 130 °C. Moreover, Mws irradiation was performed with a more efficient single-mode Mws instrument, resulting in a stronger energy density than when using a multimode domestic Mws oven, and further curing was performed in a vacuum oven. The authors confirmed that differences in FTIR spectra were related to variations in the DE, but not only; 1H-NMR, 13C-NMR and MALDI-TOF spectrometry highlighted that MW irradiation resulted in the activation of both primary and secondary -OH groups in glycerol, leading to the formation of branched pre-polymers, whereas for conventional pre-polymerisation, the formation of linear chains is more favoured. This and the faster reaction rate obtained with Mws irradiation are responsible for an overall higher DE in pre-PGS and result in shorter curing time, if compared with conventional synthesis. In fact, the authors reported that branched pre-PGS molecules are advantageous in significantly reducing the curing time but are also related to an increase in the stiffness of the final elastomer [56]. Thanks to these observations, a better understanding of the effect of Mws irradiation in the synthesis of PGS is available, allowing the fine tuning of its mechanical and biodegradation properties.

Another interesting Mws approach in PGS synthesis was reported by Lee and co-workers, who applied Mws irradiation to crosslink classically synthetised pre-PGS to obtain highly porous, tubular scaffolds for tissue engineering. Conventionally synthetised pre-PGS was dissolved in THF and loaded into a tubular mould, composed of fused sodium chloride (NaCl) granules. After THF evaporation, curing was performed in a closed vial filled with a desiccant and using an Mws reactor; heating was performed continuously for 3 h, testing different temperatures (from 120 to 170 °C). After crosslinking, NaCl porogen was removed through solvent leaching in water and the scaffold was lyophilised for further analysis. Similar to Mws pre-polymerisation, Mws curing had a faster reaction rate than conventional vacuum oven-driven crosslinking. FTIR and DSC spectra showed that after 3 h at 160 °C in MW conditions, the complete reticulation of PGS was performed, obtaining scaffolds with comparable morphology, mechanical properties, and biocompatibility as PGS cured under vacuum conditions at 150 °C for 24 h [57].

Summing up, all authors agreed with the fact that microwave irradiation tremendously increases the polymerisation rate of PGS, having undeniably positive effects on operation simplicity, time and energy saving. The use of single-mode microwave reactors resulted in more reproducible and controllable synthesis, with the possibility of fine tuning the DE and obtaining the desired mechanical and degradability properties. On the other hand, the possible use of solvents and catalysts inevitably raises questions about their presence in the polymer with potential cytotoxic substances, which could compromise PGS application in regenerative medicine, even if their use improved synthesis reproducibility. Hence, further optimisation of this synthetic technique represents an important step for the application of PGS in different research field, including regenerative medicine.

### 3.2. Ring-Opening Polymerisation

Contrary to step-growth polymerisation, chain–growth polymerisation involves the sequential and repeated addition of monomers through the presence of activated species or active centres, generated by the presence of an initiator group. The advantage of chain–growth polymerisation is the high reaction rate, quickly gaining mainly linear polymers with high molecular weight up to 10,000 kDa and a low polydispersity index. Nevertheless, branching and crosslinking is still possible, and active initiator mojeties in the bulk polymer can still lead to the deterioration of physical–chemical properties and affect biocompatibility. Chain–growth polymerizations typically involve initiation, propagation, and termination steps, with the propagation phase being driven by the products of the initiation step. Hence, as in a cascade reaction, at each addition step, a new active centre is formed and further elongation of the polymeric chain can carry on. The growth of the polymer stops either by termination or transfer steps. Termination leads to the irreversible loss of the active centre, while transfer results in the initiation of a new chain while the original one is terminated. This transfer can occur to another molecule, such as a solvent or initiator. Subsequent reactivation in a transfer step may result in branched or crosslinked products. Monomers suitable for chain–growth polymerizations usually contain double or triple bonds or possess a high ring strain. The chemical nature of the active centre determines the type of chain–growth reaction, which can be radicalic, ionic (anionic or cationic), or transition-metal mediated (coordinative or insertion). Moreover, the polymerization behaviour of a monomer depends on its structure, substitution pattern, and external factors such as the solvent, temperature, and pressure [9,67]. Ring-opening polymerisation (ROP) is comparable to chain–growth polymerisation, in which a polymer chain has a reactive end-group that reacts with a cyclic monomer, causing the opening of the ring system and the elongation of the chain. The reactive centre on the growing polymer chain can be ionic or radical. The driving force leading to ring opening is steric repulsion or bond-angle strain within the cyclic monomer due to the presence of double bonds or heteroatoms, such as oxygen. Polyesters such as poly-lactic acid and polycaprolactone are mainly produced by ROP starting from their equivalent lactones, since this technique allows to produce polymers with a well-defined chemical structure in a one-step reaction without small-molecule by-products, in large quantities, and in a relatively short time. Moreover, ROP is a fundamental technique applied to incorporate specific functional groups into biopolymers’ backbones, without requiring protection/deprotection procedures and using mild reaction conditions [68,69].

You and co-workers first reported an analogue of PGS obtained by epoxide ring-opening polymerisation, namely polysebacoyl diglyceride (PSeD, Figure 3). The epoxide-containing monomer diglycidyl sebacate was obtained by reacting sebacoyl chloride and glycidol epoxide in toluene and triethyalamine at −15 °C and under N_2_ for 30 min. After six more hours of stirring, the product was filtered and purified by flash chromatography. To obtain a PSeD polymer, an equimolar amount of diglycidyl sebacate and sebacic acid were reacted in dioxane in the presence of 0.1 mol% tetrabutylammonium bromide under N_2_, at 95 °C and for 174 h. The product was further purified by precipitation in ethyl ether. The formation of PSeD was confirmed by FTIR analysis, highlighting the presence of ester bonds in the polymer’s backbone and of free hydroxyl groups. H- and C-NMR spectra revealed the complete reaction of epoxide functionalities, while C DEPT 135 and H-C HECTOR NMR experiments were carried out to confirm the linearity of the PSeD polymer. The C DEPT 135 NMR spectrum of PSeD was used to distinguish the different carbons (CH_2_ and CH) and to discriminate CH groups bonded to free OH groups or involved in ester bonds; the H–C HETCOR NMR spectrum revealed the correlation of protons and carbons in PSeD, showing that most of the proton signals of the glyceryl moiety appeared at 4.00 ppm, while at 5.09 ppm, only weak signals associated with ester CH group were found. Hence, most of the –OH groups remained free, indicating a linear diglyceride structure. The authors estimated that about 90% of PSeD is linear and only 10% is in a branched form, while, in the case of PGS, only 45% of the polymer was reported to be linear. In optimal synthetic conditions, a PSeD polymer with a Mn of 16.6 kDa and a low PDI of 2.5 was obtained. Higher catalyst concentrations (1.4 and 3.4 %mol) led to higher PDI or even gelation due to the complete crosslinking of the polymer. Moreover, the obtained polymer was soluble in most common polar organic solvents, including dichloromethane, THF, dioxane, acetone, methanol, ethanol, DMF, and ethyl acetate. PSeD had similar thermal behaviour to PGS, with Tg lower than 100 °C, but higher Tm and Tc due to its linear structure. To obtain elastomeric, crosslinked PSeD, a mixture of PSeD and 1.1% wt. sebacic acid was cured at 120 °C for 20 h, followed by 21 h at 1.1 Torr. Mechanical tensile testing revealed a Young modulus of 1.57 ± 0.48 MPa, an ultimate tensile strength of 1.83 ± 0.06 MPa and an elongation at break of 409 ± 29%. Moreover, cyclic tensile testing showed a limited hysteresis phenomenon [70]. In a related work by Chen and co-workers, a reduction to 12 h of the curing time gained PSeD elastomers with Young modulus of 0.57 ± 0.02 MPa and an elongation at break of 136 ± 09%, indicating that the curing time of PGS influences the mechanical properties of the elastomer, showing softer elastic properties [71]. 

The presented synthetic approach is general and can be applied also to synthesize other glycerol-based linear polyesters, as shown in other work [68]. The advantage of using epoxide ROP is the possibility to obtain a PGS analogue with a well-defined linear structure and relatively high molecular weight; the low branching makes the polymer suitable for further functionalisation. On the other hand, for both pre-polymerisation and curing, inert atmosphere, extensive reaction time, and temperatures higher or about 100 °C are still needed. Moreover, similar to PGS, the curing time directly affects the degree of the crosslinking of the elastomer, influencing its mechanical properties, even though a comprehensive study on the curing time and the effects of temperature on the elastomer’s properties is not reported.

### 3.3. Enzyme-Catalysed Synthesis

Enzymatic polymerisation is an emerging green synthetic pathway for biobased, biodegradable materials. It takes advantage of the use of immobilised enzymes, the possibility to operate in mild conditions and a few side products. The advantage in using immobilized enzymes is that they can be collected and recycled, as in the case of metallic/organometallic catalysts. Nevertheless, the difficulties related to the development of suitable immobilization techniques and to the obtainment of sufficiently active immobilised enzymes should not be underestimated. Hydrolases, transferases, and oxidoreductases are the main enzymatic classes used in polymer synthesis [72]; especially in the synthesis of aliphatic polyesters, lipases (also known as esterases) have been widely employed. It is well known that in an aqueous environment, lipase catalyses the rupture of the ester bonds of insoluble triglycerides, but in non-aqueous media, other types of reactions can be catalysed, such as esterification, amidation, aminolysis, aldol condensation and Michael addition. In fact, lipase conserves a high reactivity in organic solvents such as benzene, toluene, diphenyl ether, and n-hexane, possessing a logP greater than 1.9 [73]. Recently, the use of supercritical carbon dioxide, ionic liquids, and hydrofluorocarbon solvents has been proposed as an alternative to toxic organic solvents [74]. Lipase belongs to the α/β hydrolase family, and their active site, consisting of a catalytic triad composed of serine, histidine and aspartic or glutamic acid, is highly conserved among different species. The most popular hydrolase used for the biocatalysis of polymers is *Candida antarctica* lipase B, also known as CALB; the steric hinderance at the active site limits side reactions and allows for the production of nearly linear polyesters [73]. A proper immobilisation of the enzyme is fundamental not only to allow for the recovery and reuse of the biocatalyst, but also to protect the enzyme against the solvent, temperature, pH, and to preserve its activity. Immobilisation strategies involve physical adsorption on hydrophobic supports, entrapments in microgels, and covalent bonding to substrates. Several commercial formulations of immobilised CALB are available on the market (Novozym^®^435, Novozymes A/S Krogshøjvej, Bagsværd, Denmark; Chirazyme^®^L-2, Roche molecular Biochemicals, Mannheim, Germany; LCAHNHE and LCAME, SPRIN S.p.A, Milan, Italy; etc.) and new types of reactors have been proposed for the batch production of bio-catalysed polymers, such as variable-volume view reactors and packed bed reactors [72,74].

The synthesis of PGS through enzymatic catalysis was performed only recently, resulting from other studies on the biocatalysis of the polycondensation of glycerol with diacids or diesters [75,76,77]. Godinho et al. were the first researchers who specifically reported the synthesis of PGS using both immobilized CALB (Novozym 435, N435) and a free enzyme. Glycerol and sebacic acid (1:1 mol/mol) were mixed with 100 mL of t-butanol and stirred at 250 rpm for 30 min at 60 °C before the catalyst was added; 200 μg of N435 or 500 μL of an aqueous solution of CALB were used and 10 g of zeolite was added as a desiccant. The reaction was performed at 60 °C for 24 h under stirring and in a closed reaction vessel. After catalysis, the particulate (zeolite and N435) was removed by filtration and t-butanol evaporated. Pre-PGS was washed with distilled water, centrifuged at 2000 rpm for 5 min and further decanted and desiccated in an oven at 105 °C for 2 h to remove residual water. Free CALB was not removed from the bulk polymer. The physical state of the obtained pre-PGS (viscous liquid) and the consumption of free carboxylic acid groups matched the esterification degree data reported in the literature [54]. The formation of pre-PGS was confirmed by 1H-NMR and MALDI-TOF/TOF spectroscopy; the most abundant oligomer was the linear tetramer GSGS and, in general, linear oligomers were the most abundant chemical species, even though cyclic and branched oligomers were detected. In particular, cyclic tetramer, pentamers, hexamers and heptamers were present, whereas multibranched or hyperbranched oligomers were not detected. The longest oligomers detected were tridecamers, and molecular weights smaller than 1 kDa were obtained [61].

Further investigations on the polycondensation kinetic and on the branching behaviour of CALB-catalysed PGS synthesis were performed by Perin and Felisberti; 1:1 molar amounts of glycerol and sebacic acid were reacted in various solvents (THF, acetone, t-butanol, and acetonitrile) in the presence of 13.6% *w*/*w* of acrylic resin-CALB and 5 Å molecular sieves. The reaction mixture was stirred at 150 rpm and heated for 24 h at different temperatures (30, 40, 50, 60, 70 °C). Afterwards, CALB and molecular sieves were removed by filtration, PGS was washed with acetone, and the solvent was removed under vacuum conditions. Pre-PGS with a higher degree of conversion and higher molecular weight was obtained in acetone at 40 °C. The reduction in CALB activity was found to be related to temperature (>50 °C) rather than to the solvent logP, suggesting the possibility of enzyme denaturation at high temperatures. The pre-PGS structure was studied with different 1D and 2D NMR techniques (1H-NMR, 13C-NMR, 2D-COSY, 2D-HSQC) which confirmed that a branched polymer was obtained. The polymer was composed of the following: (i) terminal glyceridic units with only one -OH group involved in the ester bond and two free -OH groups (1T, 1-monoacylglyceride, or 2T, 2-monoacylglyceride); (ii) linear repetitive units of 1,3-diacylglyceride (1,3L), or 1,2-diacylglyceride (1,2L); and (iii) dendritic/branched repetitive units of 1,2,3-triacylglyceride (1,2,3D). The most abundant units were 1T and 1,3L, confirming the higher selectivity of CALB for primary hydroxyl groups. Kinetic studies established that the increase in molecular weight is mainly attributable to CALB catalysis, whereas acyl migration is only responsible for changes of the ester position from primary to secondary hydroxyl groups, leading to branching. Actually, the synthesis of PGS exploiting biocatalysis with immobilized CALB in a non-hazardous solvent 5-fold increases the reaction rate if compared to conventional organic synthesis [43]. The authors suggested that branching via acyl migration could be controlled by changing the temperature, solvent, CALB amount, sebacic acid/glycerol feed, and with the use of activated esters. Moreover, dynamic light scattering (DLS) showed that, above a critical aggregation concentration of 0.74 mg/mL, PGS forms aggregates, reducing the availability of functional groups to further proceed with esterification [63].

The possibility to obtain low-branched and higher molecular weight PGS compared to conventional synthesis using CALB catalysis was confirmed also by Ning and Lang et al. A complex experimental design was proposed; first, equimolar amounts of sebacic acid and glycerol were heated at 150 °C for 1 h under stirring and N_2_ in order to melt the sebacic acid and obtain small oligomers that are liquid at 90 °C, obtaining a homogeneous monophasic mixture compatible with the study of CALB catalysis in neat conditions (no solvent). Afterwards, 24 h heating at 120 °C under N_2_ was performed before applying the two distinct synthetic methods. The conventional synthesis of pre-PGS was performed by lowering the pressure to 2.5 Torr and conducting the reaction at 120 °C for 72 h. CALB-catalysed synthesis was performed, reducing the temperature of the pre-treated reactant mixture to 90 °C, adding 10% *w*/*w* of N435 and stirring for 2 h under N_2_, before lowering the pressure to 100 Torr for 4 h, then to 75 Torr for 12 h, to 50 Torr for another 12 h, and finally to 25 Torr for 67 h (total enzymatic reaction time: 96 h). The esterification of glycerol and sebacic acid was followed overtime with 1H-NMR, ESI-MS, and GPC. After the first heating step (150 °C, 1 h, N_2_), abundant non-reacted sebacic acid (49.2%) and fewer dimers and trimers (35.0% and 11.8%, respectively) were detected; by prolonging the heating (120 °C, 24 h; N_2_), a slight increase in molecular weight occurred, with the formation of pentamers, hexamers and heptamers. After 12 h, a plateau was reached and sebacic acid consumption stabilized, in accordance with the already known necessity to apply a vacuum to remove water by-products, to allow for further esterification of PGS [13]. During conventional synthesis, between 55 and 67 h time point gelation occurred due to crosslinking between polymeric chains, and the reaction was stopped; an MW value of 13.8 kDa and an Mn value of 2.6 kDa were reported. On the other hand, the enzyme-catalysed reaction did not show gelation and, at 55 h time point, MW was equal to 43 kDa and Mn to 5.5 kDa, demonstrating that the high specificity of the enzyme for primary hydroxyl groups due to the steric hinderance at the active site reduced branching and, as a consequence, also gelation, especially when compared with the conventional thermal synthesis of pre-PGS. Hence, the enzymatic reaction was stopped after 96 h, when a MW of 59.4 kDa and a Mn of 6 kDa were reached. The high molecular weight of the obtained PGS led to an increase in viscosity, limiting reactant diffusivity and access to the enzyme’s active pocket, but the authors hypothesized that by using an optimized mixing shaft and an overhead mixer, these drawbacks can be overcome, and higher molecular weights can be collected. Using a peak deconvoluting mathematical method, overlayed GPC peaks were resolved in two separated peaks, describing high- and low-molecular weight fractions (HMWF and LMWF, respectively). CALB-catalysed synthesis yielded an abundant HMWF (67.6% of the AUC) with an MW of 42.8 kDa and an Mn of 25.2 kDa (PDI 1.7), whereas, with conventional synthesis, few HMWF were recovered (23.5% of the AUC) with an MW of 17.4 kDa and an Mn of 15.2 kDa (PDI 1.1); LMWFs were 32.4% and 76.5% of the AUC, MW was 3.9 and 3.6 kDa, and Mn was 2.3 and 2.5 kDa, for enzyme and conventional synthesis, respectively. These results confirmed that enzymatic synthesis is superior to conventional synthesis, yielding PGS with a greater high molecular weight fraction, only a minimal amount of branched chains, and with overall longer polymeric chains [62].

### 3.4. Ultraviolet Light-Driven Photopolymerization

Photopolymerization is a growing research field in polymer science and technology with application in industry, electronics, energy storage, and in medical and pharmaceutical areas. It is based on the conversion of luminous energy into chemical energy, used to drive organic reactions [78]. A photopolymerization system is composed of a monomer/oligomer dissolved in a proper solvent and a photoinitiator; the latter is a chemical entity that adsorbs light at a specific wavelength and transforms itself in a reactive species, responsible for the initiation of the polymerisation reaction [79]. The photoinitiator is characterized by a high quantum yield and a relative extinction coefficient similar to the output of the lamp; it can be activated by different radiation sources, from gamma and X-rays to UV/visible light, and the obtained reactive species can be a radical, cation or anion, defining the reaction mechanism of photopolymerization. The most common UV-driven photopolymerization reaction mechanisms are free-radical photopolymerization, thiol-ene photocrosslinking, and photo-mediated redox crosslinking [80,81]. Since photoinitiators have a relatively high cost and their residuals in the bulk polymer can cause deterioration of biomaterials and be potentially cytotoxic, photoinitiator-free polymerization systems have been explored. In this case, short-wavelength self-initiating monomers are used and directly incorporated in the polymeric chain; examples include electron donor/acceptor and thiol-ene systems [78].

Several authors have focused on the photopolymerization of PGS using different photopolymerization systems, reported in Figure 3. A classic approach is the application of the free-radical polymerisation of pendant ene-groups, such as acrylate, methacrylate, and fumarate systems; upon initiation, during elongation, step radicals propagate through unreacted double bonds in monomers and/or already formed oligomers, leading to the formation of the polymeric chain. The elongation of the monomer chain ends when two radical chain ends are combined with each other or react with oxygen, or by termination through chain transfer reaction [80].

Nijst and co-workers primarily investigated the acrylation of the classically synthetized PGS pre-polymer; a dichloromethane solution of pre-PGS containing 4-(dimethyl amino) pyridine (DMAP) as the catalyst was cooled to 0 °C under nitrogen (N_2_), before acryloyl chloride and an equimolar amount of triethylamine (TEA) were added. The solution was brought to room temperature and stirred for 24 h, before unreacted species were removed by dissolving the mixture in ethyl acetate, removing the TEA salts by filtration, and evaporating the organic solvent under vacuum (45 °C, 5 Pa). Through the careful evaluation of 1H-NMR spectra, the amount of acryloyl chloride incorporated in the polymer chain was calculated to be 66% of the added reactant. The acrylation involved the free secondary hydroxyl group on glycerol residues, allowing for further crosslinking between adjacent polymeric chains. The degree of acrylation was controlled by varying the amount of added acryloyl chloride and did not affect the average molecular weight of the obtained PGS-acrylated (PGS-A) macromer. To initiate photopolymerization, 0.1% *w*/*w* of 2,2-dimethoxy-2-phenylacetophenone (DMPA) was added to PGS-A, followed by UV light irradiation for 10 min (4 mW/cm^2^, 365 nm). PGS-A showed typical FTIR absorption bands for esters and hydroxyl groups and a glass transition temperature (T_g_) between −32.2 and −31.1 °C, both similar to thermally cured PGS. Mechanical properties showed to be potentially finely tuned by controlling the degree of acrylation (DA); by increasing the DA, the crosslink density increased, resulting in higher Young modulus, higher ultimate tensile strength, and lower elongation at break. Moreover, PGS-A showed to be less susceptible to esterase and hydrolytic degradation than thermally cured PGS [82]. Ifkovits and co-workers further studied the possibility of tailoring PGS elastomer properties using PGS-A. The reaction rate of the photo- and redox-initiated crosslinking of PGS-A was compared, demonstrating the advantages of photopolymerization in terms of time and conversion rate. Moreover, the possibility to tailor the gelation time by controlling the amount of the added photoinitiator was confirmed. The authors also highlighted that mechanical and biodegradation properties do not only depend on the DA, and hence on the crosslinking density, but also on the molecular weight of pre-PGS; an increase in both properties resulted in enhanced elastomeric features, affecting the Young modulus and elongation at break, biodegradability and swelling [83]. Hence, since the molecular weight of pre-PGS is highly influenced by slight variation during pre-polymerisation, reproducible synthetic pathways should be employed to avoid drastic alteration of the elastomeric properties of PGS elastomer [83,84]. On the other hand, the versatility and possible application to different soft tissue engineering fields and processing techniques are confirmed once more [38,85,86,87].

More recently, the methacrylation of PGS was proposed to overcome the shortcomings related to the synthesis of PGS-A, namely the production of toxic chlorine and the spontaneous crosslinking causing undesired side-products due to the high reactivity of acrylate groups. Two different methacrylation approaches are presented in the literature, based on two distinct reaction conditions. A classic synthetic pathway was adopted by Pashneh-Tala and co-workers; briefly, pre-PGS and 1 mg/g of 4-methoxyphenol were dissolved in dichloromethane (DCM), and equimolar amounts of triethylamine (TEA) and methacrylic anhydride were slowly added. Even in this case, the amount of added methacrylic anhydride was calculated by considering that only one secondary hydroxyl group is free to bind, and by taking into account the desired degree of methacrylation (DM). The reaction was performed at 0 °C and allowed to raise to room temperature over 24 h. PGS-methacrylated macromer (PGS-M) was collected after washing with 30 mM HCl, drying with CaCl_2_, and removing DCM at reduced pressure. Photopolymerization was achieved by adding a photoinitiator 50/50 blend of diphenyl(2,4,6-trimethylbenzoyl) phosphine oxide and 2-hydroxy-2-methypropiophenone at a concentration of 1% *w*/*w* and by UV-A irradiating for 10 min. As for PGS-A, an increase in DM resulted in enhanced elastomeric properties of the crosslinked polymer, with higher Young modulus and ultimate tensile strength, and a slower degradation rate, even in enzymatic conditions, probably due to the presence of the ester-protecting methyl group. Interestingly, no influence of the pre-PGS MW was observed for these properties, as was the case for PGS-A. Controlling the DM allowed for the tailoring of the crosslinking degree and, therefore, of the final elastomeric properties [88].

A different synthetic approach was applied by Wang et al., allowing for the use of milder reaction conditions; 2-isocyanoethyl methacrylate was added to pre-PGS (0.5:1 molar ratio) and dissolved in dimethylformamide (DMF). The mixture was heated under N_2_ to 80 °C for 20 min and quenched with deionized water. Isocyanoethyl methacrylate PGS (PGS-IM) was finally purified by precipitation. A dual-crosslinking approach was proposed; UV crosslinking was achieved by adding 1% *w*/*w* of Irgacure 2959 [2-hydroxy-4’-(2-hydroxyethoxy)-2-methylpropiophenone] and exposing the polymer to a 365 nm wavelength for 10 min at room temperature. On the other hand, free secondary -OH groups were still available due to incomplete methacrylation, and thermal curing at 150 °C in a vacuum oven (1 Torr) for 12 h was possible. A combination of both curing methods was also performed, starting with UV curing and completing the crosslinking with further thermal curing. The integration of 1H-NMR spectra showed that 32% of hydroxyl groups were successfully functionalized with pendant isocyanoethyl methacrylate groups, supporting the possibility to perform double curing. Even if the crosslinking degree was not calculated, the authors concluded that the dual-cured PGS-IM had a higher degree of esterification based on the lower swelling degree, higher glass transition temperature and higher Young modulus than UV- or thermal-cured PGS-IM. The advantage of using a dual-curing approach relied on achieving better mechanical properties and maintaining good scaffold porosity, which usually decreases during thermal curing and obviously reduces the curing time [89].

Bodakhe and co-workers investigated the synthesis of PGS fumarate (PGS-F) with the purpose to obtain an injectable and in situ crosslinkeable, yet strong, nanocomposite for bone regeneration application. Fumarylation was performed adding fumaryl chloride at different ratios to a pre-PGS solution in anhydrous DCM by using potassium carbonate (K_2_CO_3_) as an acid scavenger instead of TEA to avoid the formation of coloured compounds that could block UV irradiation to the bulk polymer. The mixture was heated and refluxed at 50 °C for 24 h. Unreacted K_2_CO_3_ and potassium chloride by-products were removed by centrifugation at 5000 rpm for 15 min, and the PGS-F was collected after removing DCM under vacuum conditions. Even in this case, to induce UV crosslinking, less cytotoxic Irgacure 2959 was used in combination with bisacyl-phosphinoxide (BAPO), added at 1% and 2% *w*/*w* of PGS-F, respectively, and photopolymerized under 315–380 nm for 15 min. The degree of fumarylation (DF) ranged between 11% and 44% and influenced the resulting crosslinking density, and hence, also the swelling degree of the PGS-F elastomer, as reported for other PGS derivatives [89]. More interestingly, if compared to PGS-A, at a given fumarylation/acrylation degree, higher mechanical strength was obtained with PGS-F [82] while still maintaining appropriate rheological properties, allowing for its use as an injectable polymer and to be applied in in situ polymerisation. This is probably attributable to the bifunctional nature of the fumarate molecule, which can bind to two PGS residues at one time, increasing molecular stiffness [58].

Yeh and co-workers focused on the development of PGS-norbornene (PGS-N), introducing, for the first time, the possibility to employ a photoinitiated thiol-ene click reaction for the crosslinking of PGS. In thiol-ene reactions, which can be performed in ambient condition, polymer chains’ growth occurs predominantly via a radical step-growth mechanism, thus allowing for the obtainment of more uniform, controlled and less susceptible to oxygen inhibition polymer networks. Moreover, higher cytocompatibility can be expected because there is no risk of the development of radical oxygen species (ROSs) such as those in radical polymerisation, and fewer amounts of photoinitiator and doses of UV irradiation are required [80]. 5-Norbornene-2-carbonyl chloride was synthetized in-house, solubilizing 4.47 mmol of 5-norbornene-2-caboxylic acid and 7.13 mmol of oxalic chloride in DCM, and adding DMF as the catalyst. After stirring for 6 h at room temperature and under N_2_, 5-norbornene-2-carbonyl chloride was collected by removing excess oxalyl chloride and the solvent under reduced pressure. Pre-PGS was dissolved in DCM containing 500 ppm of 4-methoxyphenol and 0.1% *w*/*w* of DMAP, and reacted with dropwise-added 5-norbornene-2-carboxyl chloride and ethylamine at 0 °C under N_2_. The reaction was then stirred overnight at room temperature. After adding additional 500 ppm of 4-methoxyphenol, DCM was evaporated, the product was washed with ethyl acetate and filtered to remove TEA salts, and ethyl acetate was removed. A second wash with 10mM of the hydrochloric acid of the PGS-N solution in DCM was performed and the product was dehydrated with magnesium sulphate before DCM was removed again. To induce photocrosslinking, pentaerythritol tetrakis (3-mercaptopropionate) (PETMP; 1:1 thiol/norbornene) and DMPA (0.1% *w*/*w*) were added to PGS-N and cured under UV at 365 nm (10 mW/cm^2^). Since PETMP is a four-arm thiol-bearing molecule, four different PGS-N chains can be crosslinked, reducing the amount of employed photoinitiator; moreover, the degree of crosslinking and, therefore, the rheological, mechanical and degradation properties of cured PGS-N were controlled by both the extent of norbornene modification and the thiol/norbornene ratio. Rheological analysis showed that after only 1 min of UV irradiation, the storage (G′) and loss (G′′) modulus achieved a plateau and that higher G′ modulus, hence stiffness, was obtained using a 1:1 molar ratio of thiol/norbornene groups, obtaining an effective crosslink. A higher ratio led to a drop in G′ due to the saturation of norbornene groups caused by the excess of thiols, yielding crosslinked PGS-N with pendant -SH functions. Concerning mechanical properties, tensile testing showed a positive correlation between the PETMP amount and Young modulus, whereas by reducing PETMP and, therefore, also the crosslinking degree, a softer elastomer with higher elongation percentage up to 200% was obtained. Similar to other crosslinked elastomers, reducing the DE resulted in faster hydrolytic degradation of ester bonds in PBS pH 7.4, but the effect in enzymatic or acidic conditions was not examined [59].

Finally, the radical polymerisation without the photoinitiator of PGS was proposed by Zhu and co-workers by synthetizing a cinnamate derivative of PGS. The advantages included avoiding the use of potentially cytotoxic photoinitiators and reducing the risk to obtain non-degradable aliphatic backbones. PGS-cinnamate (PGS-C) was synthetised by dissolving dehydrated PGS and DMAP in chloroform (CHCl_3_) and adding equimolar amounts of cinnamoyl chloride (50% mol/mol of pendant -OH groups) and TEA at 0 °C. The temperature was gradually increased to room temperature over 24 h, CHCl_3_ was evaporated, and PGS-C was washed with ethyl acetate and filtered to remove TEA salts. Finally, ethyl acetate was evaporated, and PGS-C was re-crystallized in ethanol. To obtain PGS-C with a higher degree of substitution (DS), multiple serial modifications were proposed by the authors, being more favourable than simply increasing the molar ratio of cinnamoyl chloride. UV curing was performed, exposing PGS-C for 2 h to a 600 W UVB lamp. During exposure to UV light with wavelengths higher than 260 nm, dimerization occurs at the ene-groups, and a cyclobutane ring is formed by a photocycloaddition reaction. It is important to underline that this is a reversible reaction and photocleavage of the cyclobutene ring can be achieved by irradiation with light at wavelengths smaller than 260 nm. The photopolymerization reaction kinetic is relatively slow if compared to other free-radical diene photopolymerizations (e.g., PGS-A or PGS-M), but this could be useful to better control and tune the mechanical properties of the elastomer. In fact, an increase in the DS of PGS-C (from 26% to 45%), and, therefore, also of crosslink density (from 6% to 20%), led to higher Young modulus. If compared to other PGS pre-polymers, such as simple pre-PGS and pre-PGS-A [13,82], higher T_g_ were found due to the presence of bulky aromatic groups. Moreover, the steric hinderance of cinnamate groups was also related to the relatively lower crosslinking density in highly substituted PGS-C; normally, with an increase in the DS in diene-functionalized polymers, a sharp increase in crosslink density and a decrease in degradation rate are noticed [85,88], but in the case of PGS-C, a high DS in hyperbranched networks is unfavourable for intramolecular dimerization. It was also hypothesised that the higher rate of degradation could be related to a secondary macroscopic network that permits rapid water diffusion in the bulk polymer, being potentially advantageous for cell adhesion and proliferation. Moreover, the authors highlighted that unlike other PGS derivatives, crosslinked PGS-C can be sterilized by autoclaving with only minimal change in the Young modulus, making it a strong candidate for in vivo applications [60].

### 3.5. Hybrid Chemical-Physical Crosslinking

With PGS being a thermoset polymer, its elasticity relies on the presence of covalent crosslinks between adjacent random coiled polymer chains. Due to the presence of free hydroxyl groups, hydrogen bonds also form in the bulk polymer, but their contribution to PGS’ elasticity is weak, with –OH groups being directly attached to the polymer backbone, therefore, lacking in the necessary mobility to undergo dynamic association and dissociation during elastic deformation, and being unable to efficiently dissipate mechanical stress [90]. Several authors investigated the possibility to introduce functional groups capable of forming non-covalent interactions, including hydrogen bonding, π–π stacks, and crystalline/glassy domains, obtaining PGS derivatives with thermoplastic-like properties. Thermoplastic polymers have the advantage of being processable because they are soluble in organic solvent and form viscous melts at temperatures above the T_g_, hence, they can be easily cast [8,91].

Pereira and co-workers firstly synthetised PGS–urethane (PGSU) by covalently crosslinking an isocyanate linker with pre-PGS and obtaining a 3D network consisting of both covalent crosslinks and hydrogen bonds. A PGS prepolymer was solubilized in DMF at a concentration of 10% *w*/*v* and heated to 55 °C in the presence of 0.05% *w*/*v* stannous 2-ethyl-hexanoate catalyst [Tin (II)]. Hexamethylene diisocyanate (HDI) was added dropwise at a variable glycerol–HDI molar ratio of 1:1, 1:0.5, 1:0.3, and the mixture reacted under N_2_ for 5 h. The solution was then cast, and the solvent evaporated for 3 days at room temperature and further 2 days in a vacuum oven at 30 °C to obtain non-porous films. The PGSU polymer proved insoluble in organic solvents, including THF, DMSO, dioxane, DMF, and DCM, due to the formation of covalent urethane bonds among HDMI residues and PGS’ hydroxyl groups. FTIR confirmed the formation of the PGS derivative, reporting the shift of the –OH stretch from 3445 cm^−1^ to wavelengths between 3359 and 3329 cm^−1^, corresponding to the –NH stretch, and showing amide I and amide II bands at 1630 and 1580 cm^−1^, while the absence of the isocyanate group band at 2270 cm^−1^ indicates the complete reaction of the HDMI. Moreover, the shift of the –OH stretch to the –NH stretch increased depending on the molar ration of HDMI. Mechanical testing reviled an increase in Young modulus from 0.7 MPa (1:0.3 ratio) up to 19.7 MPa (1:1 ratio), hence being correlated to increased crosslink density. The contribution of hydrogen bonds to the elastomeric properties of the PGSU was highlighted especially in the PGSU at a 1:0.3 ratio, for which Young modulus similar to conventional PGS were observed, but with a 3.5-fold increase in tensile strength and a 2-fold increase in elongation at break values. Finally, the PGSU also showed minimal hysteresis after 100 tensile cycles [92]. 

Ding and co-workers grafted tyramine amino acid to the PGS backbone, utilizing π–π stacking interactions among the adjacent phenol groups and achieving additional physical crosslinks. To improve tyramine pendant group mobility, pre-PGS was first bonded to a spacer, namely succinate. In total, 25% mol of succinic anhydride was reacted with pre-PGS in dioxane in the presence of pyridine and under N_2_ for 3 h. PGS–succinate (PGS-SA) was precipitated and dried, before proceeding with the tyramine coupling reaction using dicyclohexylcarbodiimide (DCC)/ N-hydroxysuccinimide (NHS)/DMAP, by adding 15 or 25% mol of tyramine to a DMF solution for 21 h and under N_2_. PGS-TA was soluble in organic solvents, including DMF, THF, and acetone. Higher tyramine ratios led to insoluble polymers. FTIR confirmed succinate coupling, showing unreacted secondary hydroxyl groups at 3467 cm^−1^, which turn into a broader band at 3367 cm^−1^ in PGS-TA; additionally, absorptions at 2926 and 2853 cm^−1^ correspond to methylene C–H stretching, while absorptions at 1732 and 1162 cm^−1^ arise from ester bonds of C=O and C–O stretching. Tyramine coupling in PGS-TA prepolymers was confirmed by amide I and II bands at 1655 and 1550 cm^−1^, and absorbance at 1516 and 830 cm^−1^ was attributed to tyramine aromatic C=C stretching and aromatic C–H bending. H-NMR further confirmed the chemical composition of PGS-TA, reporting succinate methylene protons at 3.35 ppm and 3.49 ppm, and chemical shifts at 6.97, 6.67, 3.18 and 2.58 ppm assigned to the aromatic protons and methylene protons from tyramine moieties, respectively. Moreover, H-NMR reviled that only 13 mol% of succinate was actually bonded in PGS-TA, and since the PGS-TA_15_ and PGS-TA_25_ presented a 17 mol% and 26 mol% of TA, respectively, tyramine molecules also bonded to terminal carboxylic acids deriving from sebacic acid residues. To obtain PGS-TA elastomers, the polymer was cured at 150 °C for 8, 16, or 25 h in a vacuum. From mechanical testing, it appears that tyramine pendant groups are able to improve the elastomeric properties of PGS. Being an elastomer, PGS-TA is able to restore from large mechanical deformations with minimal hysteresis, while strain at fracture, ultimate tensile strength and YM are not affected [90].

The introduction of additional physical crosslinks, either with the use of pendant groups or alternative chemical–physical crosslinkers, showed to be a suitable strategy to obtain elastomeric PGS. Depending on the intensity of non-covalent bonds and the mobility of the involved functional groups, the targeted modulation of tensile strength, elastic modulus, and overall elongation can be achieved. In fact, non-covalent crosslinks contribute to the overall crosslink density, while their dynamic association and dissociation can efficiently dissipate strain energy during elastic deformation and promote shape recoil. Nevertheless, in thermoplastic elastomers, the contribution of physical crosslinks remains a minority with respect to chemical ones; the introduction of stronger non-covalent interaction opens up to a new crosslinking strategy, named supramolecular crosslinking, described below.

### 3.6. Supramolecular Crosslinking

Supramolecular elastomers are a type of soft polymeric material that possesses both transient and (semi)permanent crosslinks. In a supramolecular polymer, the monomeric units are held together by reversible interaction; in some cases, a small polymer can be identified as a monomeric unit itself and self-assemble into a complex polymeric array. Transient and (semi)permanent crosslinks possess a different relaxation time, meaning that after an external disturbance is applied to the polymer, i.e., a deformation, a certain amount of time is needed to return to the equilibrium. Transient crosslinks usually consist of non-covalent bonds such as hydrogen bonds, π–π stacking interactions, ionic interactions, and metal–ligand coordination bonds, and are characterized by a limited relaxation time. On the other hand, (semi)permanent crosslinks such as covalent bonds, crystalline segments, and glassy hard domains have an unlimited relaxation time, implying that polymer chains do not return to their equilibrium state, or that they need an extremely long time to recoil. The combination of these molecular characteristics results in superior elastomeric properties. The difference between thermoplastic and a supramolecular elastomer relies on the ability of the latter to self-assemble into a polymeric array, thanks to intense non-covalent crosslinks. However, if the soft polymeric materials only contain supramolecular or temporary crosslinks with a limited relaxation time, they will eventually flow within a certain period and cannot function as elastomers. In simpler terms, they are just supramolecular polymers with only temporary crosslinks or bonds. Supramolecular polymer networks possess unique characteristics, including self-healing and stimuli response, as well as a significant reduction in viscosity when heated, hence being more processable [91,93]. 

Wu and co-workers firstly designed a supramolecular derivative of PGS by grafting ureido-pyrimidinone (UPy) to the polymer’s backbone (Figure 4). First UPy-HDI synthon was synthetised by reacting 0.02 mol of 2-amino-4-hydroxy-6-methylpyrimidine with 0.14 mol of HDI at 100 °C for 16 h, under N_2_. UPy-HDI was then precipitated with penthane and dried, before being reacted with the PGS prepolymer at different ratios of 2:10, 3:10, 4:10, and 5:10 in DMF and in presence of a stannous octoate catalyst at 85 °C for 12 h. PGS-graft-UPy (PGS-U) was then precipitated in diethyl ether and dried in vacuum conditions before use. FTIR and H-NMR confirmed the grafting of UPy showing, respectively, absorption bands at 1580 and 1660 cm^−1^ characteristic of amide groups, and a band at 3120 cm^−1^ from urethane–NH groups; additionally, peaks at 2.25, 11.91 and 13.14 ppm correspond to the protons of UPy segments, and the signals at 5.26 ppm arise from the urethane proton of HDI segments. To prove the hydrogen bonding formed between UPy dimers, viscosity analyses were performed, showing a large increase in viscosity, hence cohesion, when increasing the UPy ratio. Moreover, PGS-U films obtained by solvent casting showed tensile strength and YM up to 4.6 ± 0.2 MPa and 32.8 ± 0.5 MPa, respectively, while a crosslinking density up to 4413 ± 62 mol/m^3^ was calculated; higher elongation between 500% and 610% was obtained by grafting UPy at 3:10 or 4:10 ratios. In addition to adjustable mechanical characteristics based on the UPy content, PGS-U elastomers showed efficient self-healing and quick shape–memory capabilities, demonstrated to be suitable for the preparation of biomimetic scaffolds, controlled drug delivery systems, surface antibacterial composites, and cell co-culture systems [94].

Similar work was performed by Chen and co-workers, who incorporated UPy pendant groups into PSeD, with the latter synthetised as reported by You et al. [70]. The reaction conditions used to obtain UPy-HDI and graft it to PSeD are comparable to the aforementioned method described by Wu et al. [94], but with PSeD-U being a linear polymer, an additional curing step at 130 °C in a vacuum oven, for 6, 12, and 24 h, was performed in order to introduce the covalent crosslinks necessary to obtain a supramolecular elastomer, rather than a simple supramolecular polymer. UPy-HDI was incorporated at a 1:10, 2:10 and 3:10 ratio with PSeD, obtaining a wide range of mechanical properties. Rheological analysis showed that the covalent crosslinks had an influence on the storage modulus, hence, the toughness of the polymer, while non-covalent hydrogen bond crosslinks had a more dominant effect on the loss modulus, hence, the viscosity of the polymer. In fact, when the rotational frequency approached the hydrogen bond kinetics, dissociations of the bonds and chain relaxation led to energy dissipation. This behaviour was observed also in mechanical testing; under small strain, the hydrogen bonds in the elastomer remained mechanically undetectable, responding minimally to external stress, and thus, having only a small impact on the elastomer’s modulus. As the strain increased, the hydrogen bonds became more responsive and promoted chain relaxation, dissipating energy, and thereby, enhancing the elastomer’s strength and toughness. When compared to PSeD elastomers, PSeD-U elastomers showed a three-fold increase in strength and an 11-fold increase in toughness, all while maintaining a low modulus (0.64 MPa). The observed non-linear mechanical properties of PSeD-U polymers make ideal candidates for soft tissue applications [71].

In conclusion, supramolecular PGS-base polymers are a powerful and versatile platform with highly tuneable mechanical properties depending on the UPy content and on the ratio between hydrogen and covalent crosslinks. Moreover, by taking advantage of the self-assembly properties, polymers with outstanding dynamic features can be obtained for application in various biomedical applications.

### 3.7. PGS Co-Polymers

With the aim to specifically regulate the mechanical properties of PGS, rather than its pre-polymerisation and crosslinking method, a variety of PGS co-polymers are reported in the literature. Typically, the crosslinking densities of polymers are adjusted by modifying the molecular weights of their pre-condensed form. However, this method can cause physical entanglements and chain extension in some polymers when they are crosslinked through radical polymerization, which may restrict their use in subsequent processing steps. A different strategy involves the polycondensation of PGS with linear polymers using a two-step polycondensation process; first, the linear modifier polymer and sebacic acid are polymerized, followed by the introduction of glycerol. By varying the content of the modifier polymer and the proportion of sebacic acid to glycerol, a range of PGS-co-polymers can be obtained [42]. Table 1 summarizes the mechanical properties (elastic modulus, ultimate tensile strength, and maximum elongation) of PGS-co-polymers.

## 4. Polymer Characterisation 

Proper characterisation methods are fundamental when performing polymer synthesis. Molecular structure, the nature of chemical bonding, thermal behaviour, the degree of substitution and consequent molecular weight are specific properties that identify a polymer. For PGS, the preferred characterizing techniques reported in literature, and used to follow the progress of the synthesis and to study the influence of different parameters on the final product, are Fourier-Transform InfraRed (FTIR) spectroscopy, Nuclear Magnetic Resonance (NMR) and Mass Spectroscopy (MS). FTIR allows for monitoring the esterification reaction in both pre-polymer and crosslinked elastomer thanks to the shift of the carbonyl signal from 1682 cm^−1^ (characteristic of C=O in free carboxyl acid groups) to 1735 cm^−1^ (specific for C=O imply in an ester bond). NMR analysis shows to be exhaustive in the determination of the five different acyl-glycerides obtained during pre-polymerisation (Figure 1), which show a slightly different chemical shift in 1H NMR and, more clearly, in 13C NMR. Furthermore, this technique allows for the determination of the sebacic acid and glycerol ratio in the pre-polymer and the quantification of the degree of functionalization in UV-reactive PGS derivatives (Figure 2). MS constitutes a valid technique to clarify the MW of the different species, in particular during prepolymer synthesis. Electrospray Ionization (ESI) represents the first choice of ionization source, with the positive mode able to detect the majority of the species and the negative mode used to appreciate sebacic acid-terminated PGS macromolecules. These techniques, along with Differential Scanning Calorimetry (DSC), acid and ester group titration and mechanical testing, contribute also to the determination of the degree of esterification and crosslinking. A summary of the characterisation technique parameters and main findings are reported in Table 2.

## 5. Polymer Tailoring

PGS polymer’s physical, chemical, mechanical, and biological characteristics can be tailored and matched to a specific application field starting from the selection of synthetic parameters, including reagent ratio, temperature, time, and reaction atmosphere. Moreover, the post-processing of the pre-polymer and cured elastomer can influence its final performance. Several articles focus on studying the process parameters to optimize PGS synthesis, producing highly variable and sometimes conflicting results. This is mainly related to the high number of parameter combinations applicable to the polymer synthesis and to the different analytical techniques that can be used to characterize the polymer itself (see Table 1). Below, an overview on the main findings regarding the influence of different synthetic parameters in PGS preparation is reported.

### 5.1. Reagents Ratio

Since the molar ratio of the reactants can influence the molecular weight and the structure of a polyester, as well as the rate of the polycondensation reaction [9], the variation of the glycerol and sebacic acid ratio was investigated. In the literature, PGS is mainly synthetized using a 1:1 molar ratio of glycerol and sebacic acid, which is close, but not equal to, the stoichiometric ratio of 3 hydroxyl groups in glycerol to 2 carboxyl groups in sebacic acid [50]. This molar ratio was the first ever ratio investigated by Wang and co-workers in 2002, and was preferred by the authors because it leads to a not fully crosslinked polymer, showing lower stiffness and higher resilience than a fully crosslinked PGS, which is obtained by reacting stoichiometric amounts (2:3 molar ratio) of glycerol and sebacic acid [13]. Since then, several authors have investigated alternative molar ratios of reactants and their influence on PGS synthesis.

Maliger and co-workers studied the reaction kinetic of 0.6 (equal to 2:3), 0.8 (equal to 2:2.5) and 1.0 (equal to 2:2) glycerol/sebacic acid ratios pre-polymerised between 120 and 140 °C until gelation at atmospheric pressure. Toluene was used as a solvent in order to slow down the reaction for kinetic studies. The authors reported a first-order kinetic for pre-PGS synthesis, until the reaction turned viscous and became diffusion driven. With an increase in the molar ratio from 0.6 to 1.0, a decrease in the threshold energy required to form activated complexes was observed, proving favourable for the reaction rate. On the other hand, the average functionalities (ƒ_AV_) calculated using Pinner’s equation decreased; ƒ_AV_ is indicative of the degree of polymerisation, which increases with higher values of ƒ_AV_. Hence, by approaching a 1.0 molar ratio, the reaction rate increased and the crosslink density decreased. As a result, a variation in the physical and mechanical properties of the elastomer occurred; softer resins and lower Young moduli were obtained by increasing the glycerol/sebacic acid molar ratio, whereas, by approaching stoichiometric ratios, a stiffer polymer with higher Young modulus was obtained [43].

These results are supported by the work of Kafouris et al. and Conejero-García et al.; both authors tested the equimolar ratio, excess of glycerol and excess of sebacic acid, and studied their influence on the molecular weight and crosslink degree of PGS. The excess of glycerol (2:1) yielded sticky and poorly crosslinked samples after curing [50]; the samples were still soluble in THF, and had a multimodal molecular weight distribution and rather low MW [44]. Equimolar amounts of reactants reached higher molecular weights, with unimodal composition but a high polydispersity index [44] and intermediate crosslink density [50]. Glycerol/sebacic acid amounts close to the stoichiometric ratio (e.g., 2:3 and 2:4) yielded elastomers already at the end of the pre-polymerisation step, and curing was not necessary; in fact, a 70% to 80% monomer-to-polymer conversion was achieved in the first hours of pre-polymerisation, and high MW with low PDI was achieved before curing. The exact stoichiometric ratio of 2:3 resulted in the stiffest of all elastomers, as a result of the highest crosslink density [44]. Finally, by increasing the reactant ratio from 2:4 to 2:5, a decrease in stiffness was observed, even though a 100% conversion of the monomers was achieved. In general, an increase in sebacic acid content resulted in an increase in sol content [44], and the authors concluded that the excess of sebacic acid led to a branched molecular structure, interfering with the actual crosslink network formation of the elastomer; this resulted in poor mechanical properties, as well as a higher sol fraction and higher hydration and degradation rate in basic media [50].

### 5.2. Reaction Temperature and Time 

Among all synthetic parameters, temperature and time are the most studied in the literature, and their tuning in pre-polymerisation and curing remains one of the main approaches for the tailoring of the PGS elastomer. Temperature has a main role in controlling the reaction rate and kinetics [37], but should also be chosen with consideration for the melting and/or boiling points of the reactants to obtain a homogeneous mixture and avoid the loss of glycerol by evaporation. A temperature between 130 and 290 °C would be optimal considering the melting point of sebacic acid and the boiling point of glycerol. If a vacuum is used, especially in the pre-polymerisation step, a reduction in glycerol’s boiling point has to be expected, and consequently, the temperature needs to be adjusted. For the pre-polymerisation step, temperatures ranging from 110 to 180 °C were tested [16,37,54,108]; lower polycondensation temperatures between 110 and 130 °C required a reaction time between 24 and 72 h, whereas higher temperatures between 150 and 180 °C seemed to reduce the reaction time between 4 to 8 h. The pre-polymerisation step normally yields polymers with a DE between 60% and almost 80% [54] when a harsh variation in the reactants’ mixture viscosity is detected [36,46]. If a vacuum is applied during the first synthetic step, an enhancement of the reaction rate can be achieved by removing water by-products but, as already mentioned, a higher loss of glycerol occurs, especially at polycondensation temperatures above 150 to 180 °C. If glycerol loss is not evaluated, the esterification degree cannot be correctly assessed. At this point, the still-soluble pre-polymer is transferred into an oven to proceed with the final curing. Pre-PGS curing can be performed again between 110 and 180 °C, but longer reaction time is needed if compared to pre-polymerisation; the secondary hydroxyl groups involved in crosslinking have a slower reaction rate [37] and to overcome this drawback, curing times between 24 and 168 h are applied, even in combination with a vacuum [36,37,50,54,103,105]. Generally speaking, an increase in crosslink density seems to be dependent especially on curing temperature rather than time due to its direct influence on secondary hydroxyl groups’ reactivity, whereas when low temperatures (110–130 °C) are applied, time becomes a crucial factor and longer curing steps are able to ensure effective crosslinking. There is no rule of thumb regarding the influence of time and temperature on the extent of crosslinking and the resulting mechanical properties, swelling behaviour, and sol content. 

### 5.3. Atmosphere Influence 

Contrary to time, temperature and the monomer ratio, atmosphere characteristics are poorly studied in PGS synthesis. As already mentioned, pre-polymerisation is usually performed in an inert atmosphere to avoid the oxidation of reactants [8,13,37], but only recently was the real need and influence of inert gasses on the polymer’s characteristics extensively examined by Martìn-Cabezuelo and co-workers. In their study, pre-PGS was synthetized by mixing 1:1 sebacic acid and glycerol, heated at 130 °C for up to 24 h, and by applying different types of atmosphere; argon (Ar) or nitrogen (N_2_) were selected as inert gasses, whereas oxygen (O_2_) was chosen as oxidative atmosphere. Dry compressed air (DA) was used as a representative for mixed-oxidative and inert atmospheres and air with 33% relative humidity (HA) was used to study the effect of aqueous vapor. Cured elastomers were obtained by curing pre-PGS at 130 °C for 48 h. No vacuum was applied in both synthetic steps. FTIR analyses were performed to calculate the −COO−/−COOH and primary/secondary−OH ratios and evaluate the extend of the pre-polymerisation reaction; inert gasses reduced the interaction between carboxylic and hydroxyl groups, resulting in a lower esterification degree when compared to DA, O_2_ and HA, which showed a growing esterification trend suggesting that oxidative atmospheres enhance the reaction rate. Moreover, water vapour in HA is thought to reduce the evaporation of glycerol, minimizing alteration in the reactant ratio, and bringing the reaction to a higher extent [102]. In fact, it was demonstrated that in binary mixtures of glycerol and water vapour, mutual depression of vapour pressure occurs, leading to a reduction in the glycerol evaporation rate and to its condensation, even if the vapour is not completely saturated [109]. Either way, the formation of an azeotrope between glycerol and water during polycondensation occurs and minimal loss of the reactant is unavoidable [37]. Martìn-Cabezuelo and co-workers also highlighted that pre-PGS obtained with oxidative atmospheres have a higher degree of branching due to O_2_ high-electron affinity, which removes electrons from monomers and enhances secondary hydroxyl groups’ reactivity. Inert gasses, and especially Ar, reduced branching and had a positive effect on cured PGS, which was more elastic and tough. On the other hand, less crosslinking was achieved by curing pre-PGS from oxidative atmospheres, resulting in a softer polymer. Hence, the atmosphere characteristics were not only influenced by the reaction rate in the pre-polymerisation step, but also by the structure of the pre-PGS and the efficiency of crosslinking in the final curing step.

### 5.4. Purification of PGS Pre-Polymer and Elastomer

Some of the advantages related to the synthesis of PGS-based elastomers are the use of small amounts of catalysts, the solvent-free conditions, and the harmless and biodegradable nature of monomers, which can be easily metabolized. Nevertheless, unreacted monomers can possibly impair biocompatibility and seem to influence PGS elastomer’s characteristics, being one of the possible reasons for conflicting and not reproducible results, as reported in the literature [50]. Therefore, some authors proposed different purification methods for both pre-PGS and the final elastomer. Pre-PGS precipitation methods in different types of non-solvents were tested. Gadomska-Gajadhur and co-workers proposed re-precipitation of 100% *w*/*v* dioxane/pre-polymer solution in 5 °C distilled water, collecting the precipitate by Buchner filtration and drying the product at 45 °C for 24 h. FTIR spectra confirmed the removal of unreacted sebacic acid form crude pre-PGS, showing a reduction in the peaks related to the acid. Moreover, a reduction in the acidic number was observed and a more accurate calculation of the degree of esterification was possible [110]. It has to be pointed out that this method could prove less effective if high amounts of unreacted sebacic acid are present, due to the low water solubility of the monomer (<1 mg/mL, source: PubChem.ncbi.nlm.nih.gov), whereas unreacted glycerol can be easily removed thanks to its high water solubility (>100 mg/mL). An interesting approach was proposed by Ning and co-workers, who applied fractional precipitation not with purification purposes but with the intent to isolate pre-PGS chains with high molecular weight and reduce the polydispersity index, obtaining a more uniform product. The authors precipitated 50% *w*/*v* pre-PGS in THF using nine parts of different alcohols, namely methanol, iso-propanol and iso-pentanol. Pre-polymer suspensions were stored at −20 °c for 16 h to facilitate precipitation, then the supernatant was removed after centrifugation and samples were fully dried in a vacuum oven at 50 °C for 24 h. Size exclusion chromatography (SEC) was used to investigate average MW, Mn and PDI. Methanol proved more effective in increasing the average MW and reducing PDI, yielding only 27% of precipitate, suggesting a high abundance of lower-molecular-weight polymer factions. Isopropanol led to the highest precipitation yield (70%) and, as a consequence, only a slight increase in MW and decrease in PDI were observed. A refinement of the fractional precipitation of pre-PGS could represent a valid strategy to standardize pre-PGS products [62]. On the other hand, in the case of cured PGS, to evaluate the crosslinking degree of the elastomer, the soaking of PGS elastomer slabs in THF or ethanol has been proposed by different authors to remove the sol content composed of unreacted monomers and uncrosslinked low MW polymer chains [16,31,50,82]. Conejero-García and co-workers pointed out that the lack of sol removal after curing and the biased calculation of the crosslinking degree could be among the main reasons for the inconsistencies in the literature regarding the correlation of crosslink density with mechanical properties, swelling behaviour and degradation in aqueous media. On the basis of the fact that crosslinked PGS is insoluble in any organic solvent, the extraction of the sol through swelling in solvents represents a valid approach for PGS elastomer purification [50]. 

## 6. Conclusions

More than twenty years of literature and a total of more than 600 articles have proven that poly (glycerol sebacate) is a biodegradable elastomer with a high potential in tissue engineering due to its peculiar mechanical properties, biodegradability and biocompatibility. This review focused on summarizing the most significant synthetic approaches leading to PGS elastomers, reporting the most appropriate techniques for polymer characterisation and the methods that can be used to obtain tailored derivatives. PGS elastomer is classically synthetized through the heat-driven polycondensation of its monomers, glycerol and sebacic acid, at high temperatures (>100 °C) and for several hours (>24 h), while applying vacuum conditions and purge gas. The elastomeric behaviour is guaranteed by the formation of crosslinks between linear polymeric chains; therefore, controlling the crosslink density of the elastomer represents a key factor in PGS synthesis. The high reactivity of the primary hydroxyls of glycerol lead to the synthesis of mainly linear chains, but the formation of branched pre-PGS is still possible. Increasing the temperature in the pre-polymerisation accelerates the reaction, reaching the gelation point earlier, but with the risk of increasing the fraction of branched polymer molecules. Moreover, these harsh synthetic parameters are not only energy and time consuming but could lead to glycerol loss and the alteration of the monomer ratio. Reactants ratio had been proven to be a crucial parameter in determining the final crosslink density and should therefore be always determined in the final elastomer. Glycerol and sebacic acid sub-stoichiometric 1:1 molar ratio remains the preferred reactant ratio because of the better control over the chemical structure and mechanical properties of the PGS elastomer; branching can be enhanced by an excess of sebacic acid and lead to completely crosslinked elastomers with excessive stiffness, while an excess of glycerol leads to soft polymers and poor mechanical properties. Moreover, a 1:1 molar ratio allows us to obtain an intermediate reaction rate and to separate the reaction itself in the pre-polymerisation step and crosslinking, thus allowing us to tune the mechanical properties by defining the desired crosslinking degree. Concerning the reaction atmosphere, several authors reported that the use of vacuum conditions at least in the pre-polymerisation step is not mandatory; oxidative atmosphere can be used to enhance the esterification degree, and especially humid air seems to have a positive effect on reducing glycerol loss. Therefore, mixed atmospheres (e.g., humid air), rather than purely oxidative or inert atmospheres, ensure a final polymer with a good esterification degree and not excessive branching [102]. Alternative synthetic techniques, including microwave-assisted polycondensation, UV polymerisation and enzymatic catalysis, have been proposed to reduce time and energy consumption. The rational of Mws technology relies on the possibility to heat reactants intensively and almost instantly, thus tremendously enhance the reactivity of glycerol and sebacic acid. The main drawbacks of these techniques are the high risk of glycerol loss in the case of overheating or the use of open reaction vessels, and the activation of secondary hydroxyl groups that could enhance branching during the pre-polymerisation step. The use of closed-reaction vessels and single-mode microwave reactors could ensure better control over glycerol evaporation and temperature, maintaining an appropriate reactant ratio and reducing branching. Moreover, the lack of the scalability of conventional Mws technology due to limited volumetric heating could be addressed by using flow chemistry approaches [111]. To avoid high temperatures in pre-PGS synthesis, and to truly reduce to a minimum the formation of branched oligomers, enzymatic approaches using lipases provide highly promising results. The selectivity of CALB for primary hydroxyls not only impedes branching, but also catalyses the reaction, allowing for the obtainment of higher molecular weight pre-PGS. The use of immobilized CALB is certainly more advantageous in terms of costs because of the possibility to recover the enzyme and recycle it. The use of free CALB raises questions about the stability of biodegradable PGS, which remains in contact with an enzyme that catalyses ester hydrolysis in aqueous media, and is therefore not recommended. The lack of published work on crosslinked enzymatic-synthesized PGS does not allow for an evaluation of the mechanical and biodegradable properties of the cured elastomer, which is likely influenced by the higher MW and the higher number of secondary hydroxyl groups for the crosslinking reaction. Moreover, even if enzyme immobilisation is suitable for industrial application, currently, the implementation of enzyme-catalysed polymerisation remains low [72]. Regarding alternative methods for crosslinking in mild conditions, cold curing by derivatising secondary hydroxyl groups with photocrosslinkeable functional groups, including diene and thiol-ene reactive moieties, is very promising. This technique particularly enhances the processability and applicability of PGS in the soft tissue engineering field. In this case, the degree of substitution with UV-curable groups directly influences the crosslinking degree, which can also be finely tuned by regulating the exposure to UV light. Acrylate and methacrylate derivatisation is a well-established method to perform crosslinking in tissue engineering scaffolds, and the use of multifunctional crosslinkers such as fumarate and norbornene-PETMP can lead to pronounced mechanical properties, even with lower degrees of derivatisations. Moreover, the introduction of additional aliphatic backbones can improve the fast degradability of the polymer. Nevertheless, the need for the purification of crosslinked PGS from photoinitiators and derivatisation reaction by-products still represents a possible issue in biocompatibility. The use of cinnamate pendant groups could represent a valid alternative to avoid the use of photoinitiators and suggests the possibility to obtain a photoresponsive elastomer, the mechanical properties of which can be changed after scaffold production by means of photo-cleavage. Alternatively to UV crosslinking, the introduction of pendant groups capable of physical crosslinking gained elastomers with outstanding elastomeric properties, especially when considering supramolecular elastomers. Self-healing and shape–memory features were obtained by grafting pyrimidinone residues with a polyurethane spacer to the backbone of PGS (and PSeD) using mild synthetic conditions. Such polymeric platforms could be further assembled with other polymers/bioactive agents as far they bear the same pendant group and are, therefore, able to coordinate and aggregate. Given the variety of hydrogen bonding motifs (i.e., double, triple, quadruple, and multiple bonding), new supramolecular derivatives of PGS could be potentially designed. Moreover, such types of polymers can be assembled with different methods, including the covalent polymerization of supramolecular functional monomers, in situ self-sorting process, and kinetically controlled living supramolecular polymerization; hence, multiple tools are available to control the synthesis of supramolecular elastomers. Nevertheless, it should be kept in mind that multiple hydrogen bonding alone is not enough for obtaining supramolecular polymers, since also other non-covalent interactions (e.g., π–π stacking, ion-dipole, metal–ligand interactions) are necessary [112]. 

Both classic and alternative methods usually do not allow for the complete conversion of the monomers, affording a final elastomer characterized by a wide variety of polymeric chains with different molecular weights. The use of fractional precipitation is one possible approach to narrow the MW and Mn of pre-PGS and obtain more standardized products. After curing, unbound oligomers should be removed from the scaffold to allow for the correct evaluation of its chemical, physical and mechanical properties. Moreover, non-crosslinked monomers and oligomers could leach from the scaffold and impair the elastomer’s biocompatibility. In fact, we believe that a combination of the mentioned alternative synthetic approaches, together with the development of effective purification methods, could promote a more reproducible and reliable production of PGS and favour its applicability in clinics. The environmental impacts associated with the synthesis of PGS and its derivatives are primarily related to energy consumption and the solvent waste generated during the process, and should be evaluated when developing new synthetic methods. Techniques such as microwave-assisted polycondensation and enzyme-catalysed synthesis are, by far, the best strategies to reduce energy consumption during the pre-polymerisation phase by reducing the reaction time and/or reaction temperature, as well as the use of an alternative mechanism for crosslinking. Nevertheless, all discussed synthetic approaches inevitably apply organic solvents (e.g., halogenated organic solvents, tetrahydrofuran, etc.) which pose risks to both human health and the environment due to their toxicity, flammability, and potential for ozone depletion. Considering the significant role of polymers in biomedical application, efforts have been conducted to explore alternative reaction mediums to avoid the use of harmful solvents and promote more eco-friendly processes. The continuous pursuit of selecting the right solvent and understanding its role in the entire chemical process has led to new technologies that use so-called green solvents. In the field of polymer chemistry, the most frequently used green solvents include ionic liquids, deep eutectic solvents, liquid polymers, supercritical carbon dioxide, and switchable solvents [113,114]. To the best of our knowledge, currently no report on the use of such green solvents for the synthesis and processing of PGS and its derivatives has been published; hence, research efforts in this direction are highly desirable.

## Data Availability

Not applicable.

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
