# Peer review of "Polyglycerol Sebacate Elastomer: A Critical Overview of Synthetic Methods and Characterisation Techniques"

_polymers, 2024, doi:10.3390/polym16101405_

Round 1

Reviewer 1 Report

Comments and Suggestions for Authors

In the manuscript titled “Polyglycerol Sebacate Elastomer: A Critical Overview on Synthetic Methods and Characterisation Techniques,” the authors provide an in-depth review of the synthesis, characterization, and applications of poly(glycerol sebacate) (PGS), an elastomeric biopolymer applied in soft tissue engineering. The review starts with the statistical results of relatively published literature, which indicates that PSG has been widely studied and highlights its potential applications. Then, the authors discuss different PGS synthesis methods and demonstrate their advantages and disadvantages, including traditional thermal polycondensation and innovative techniques such as microwave-assisted synthesis and enzymatic catalysis. Based on the physical and chemical properties of PGS, such as molecular weight, crosslinking density, and its mechanical properties like tensile strength and elasticity,  each synthesis method’s outcome is evaluated in terms of efficiency and reproducibility, assessing the quality of the resulting polymer for specific biomedical applications. In the end, The authors conclude that while traditional methods are effective, essential synthesis methods offer advantages regarding reaction speed and environmental impact but require detailed control over reaction parameters to ensure consistent product quality.

In summary, this manuscript makes a significant contribution to the field of soft tissue engineering by briefly compiling and reviewing various synthetic methods for PGS and evaluating their suitability for producing biocompatible and biodegradable polymers.

However, the following points should be addressed before publication:

1. The review summarizes existing methods without introducing new experimental data that could significantly advance the field. Furthermore, the lack of figures illustrating novel experimental results restricts the ability to compare the efficacy of the described synthetic methods.

2. A detailed assessment of the environmental impacts associated with the synthetic methods for PGS is lacking, which could limit the discussion on potential future research directions.

Comments on the Quality of English Language

The manuscript’s language and clarity could be improved to meet the journal’s standards. For example, ensure consistency in terms such as “characterisation” and “characterization.” Additionally, some sentences are overly long and complex, potentially confusing readers. Simplifying these could enhance understanding and readability.

Reviewer 2 Report

Comments and Suggestions for Authors

In this review, authors summarized different synthetic approaches of PGS, ranging from classic thermal polyesterification and curing to microwave-assisted organic synthesis, UV-crosslinking and enzymatic catalysis. Also, classic and alternative synthetic methods, characterization, and tailoring techniques are reviewed. While the manuscript is well-structured and presents fresh insights, it omits some important synthesis methods developed in recent years.  Moreover, the applications part based on these approaches is a bit weak.  Therefore, I would like to recommend a major revision.

1. Introduction: It would be beneficial to provide more recent reviews about PGS and its derivatives for the readers' better understanding. (i.e., doi.org/10.1016/j.eurpolymj.2021.110830; doi.org/10.1002/adhm.202002026)

2. Introduction; FDA already approved glycerol and sebacic acid containing polymers for medical use. It is necessary to verify the accuracy of this statement. While glycerol-based materials have FDA approval, the status of sebacic acid remains unclear. The discussion should also address the clinical potential of PGS and barriers to its clinical application.

3. The scope of the review could be broadened to include applications in various fields of tissue engineering beyond soft tissue. This could include antibacterial devices, hard tissue engineering, drug delivery. A table summarizing these applications developed in recent years is strongly recommended.

4. Synthesis and crosslinking of PGS: The synthesis approaches discussed are somewhat limited. Recent advancements such as urethane-based crosslinking; PEGylated synthesis; thiol–ene click reaction-based crosslinking; thiol-Michael addition click reaction can be included. These are examples and the authors are strongly recommended to include more recent references

5. Please also include the degradation profile test in the PGS characterization.

Comments on the Quality of English Language

Minor editing of English language required

Round 2

Reviewer 2 Report

Comments and Suggestions for Authors

The authors have addressed my concerns.